# UV-B Irradiation to Amino Acids and Carbohydrate Metabolism in *Rhododendron chrysanthum* Leaves by Coupling Deep Transcriptome and Metabolome Analysis

**DOI:** 10.3390/plants11202730

**Published:** 2022-10-15

**Authors:** Qi Sun, Meiqi Liu, Kun Cao, Hongwei Xu, Xiaofu Zhou

**Affiliations:** Jilin Provincial Key Laboratory of Plant Resource Science and Green Production, Jilin Normal University, Siping 136000, China

**Keywords:** amino acids, carbohydrates, UV-B irradiation, *Rhododendron chrysanthum*, metabolome, transcriptome

## Abstract

Under natural environmental conditions, excess UV-B stress can cause serious injuries to plants. However, domestication conditions may allow the plant to better cope with the upcoming UV-B stress. The leaves of *Rhododendron chrysanthum* are an evergreen plant that grows at low temperatures and high altitudes in the Changbai Mountains, where the harsh ecological environment gives it different UV resistance properties. Metabolites in *R. chrysanthum* have a significant impact on UV-B resistance, but there are few studies on the dynamics of their material composition and gene expression levels. We used a combination of gas chromatography time-of-flight mass spectrometry and transcriptomics to analyze domesticated and undomesticated *R. chrysanthum* under UV-B radiation. A total of 404 metabolites were identified, of which amino acids were significantly higher and carbohydrates were significantly lower in domesticated *R. chrysanthum*. Transcript profiles throughout *R. chrysanthum* under UV-B were constructed and analyzed, with an emphasis on sugar and amino acid metabolism. The transcript levels of genes associated with sucrose and starch metabolism during UV-B resistance in *R. chrysanthum* showed a consistent trend with metabolite content, while amino acid metabolism was the opposite. We used metabolomics and transcriptomics approaches to obtain dynamic changes in metabolite and gene levels during UV-B resistance in *R. chrysanthum*. These results will provide some insights to elucidate the molecular mechanisms of UV tolerance in plants.

## 1. Introduction

The increase in solar UV radiation (especially UV- B, 280–315 nm) at the Earth’s surface is due to stratospheric ozone depletion from chlorofluorocarbon and halon emissions [1]. Numerous deleterious effects have been reported on plants due to increased UV radiation levels, such as damage to DNA macromolecules, decreased photosynthetic rates, and changes in growth, development, and morphology, thus affecting plant productivity [2,3,4]. Despite the success in reducing ozone-depleting substances in recent years, the full recovery of stratospheric ozone will take decades; the ozone hole is expected to return to its 1980 level around 2070 [5].However, in the near and long term, ozone depletion remains a persistent environmental concern for agroecosystems in terms of its impact on solar UV radiation [6].

In previous studies, the metabolites of *R. chrysanthum* plant leaves exposed to UVB were identified for the first time using gas chromatography-time of flight mass spectrometry (GC-TOFMS), and sugars and amino acids have been shown to inhibit or enhance biosynthesis and degradation under UV-B radiation [7]. UV-B stress allows plants to improve their metabolism and reconfigure their internal metabolic networks in a way [8]. Under UV-B stress, plants produce large amounts of amino acids and their derivatives (l-cysteine, ketoleucine, n-acetyl- l-aspartate, l-tyrosine, homocysteine) [9]. This change in activity occurs in the primary metabolite phase after UV-B treatment and leads to improved production of functional secondary metabolites to protect plants from UV-B [9,10].

Plant tolerance to UV is controlled by multiple genes [11]. By high-throughput RNA sequencing (RNA-seq), we found that 71,651 unigenes showed different expression patterns under UV-B stress in our transcriptome study of two species of *R. chrysanthum*, and the transcript abundance of most unigenes encoding cysteine and methionine metabolic pathways as well as glycine, serine and threonine metabolic pathways was reduced under UV-B stress. However, a comprehensive understanding of the metabolic status of R. chrysanthum in combination with deep transcriptomic and metabolomic analyses is important to clear the mechanisms of *R. chrysanthum* defense against UV-B responses.

*R. chrysanthum* is an ornamental plant used in traditional medicine in Asia, North America and Europe. It is mainly used to treat inflammation, pain, skin diseases, the common cold and gastrointestinal diseases [12]. *R. chrysanthum* is a unique species in Changbai Mountain in China, the environment of Changbai Mountain is not ideal: the air temperature is low in the winter, the air is thin and the solar radiation is strong, with UV-B radiation being the main abiotic stress factor in this area. Therefore, *R. chrysanthum* is an important plant resource to study plant stress tolerance [13]. In the present study, two *R. chrysanthum* with different UV-B tolerance were used as materials to study their response to UV-B stress through experiments. The objectives of this study were (I) to perform a parallel analysis, analyze metabolites by GC-TOFMS and carry out RNA-Seq to determine genome-wide transcripts; and (II) to investigate the effects of UV-B on the regulatory patterns of amino acid and carbohydrate metabolic pathways of some candidate genes in *R. chrysanthum*.

## 2. Results

### 2.1. Analysis of Metabolites in Rhododendron Chrysanthum

In this study, the first non-targeted GC-TOFMS method was used to determine the differential metabolites of *R. chrysanthum* plants under UV-B treatment. In this experiment, a total of 404 peaks were detected using a metabolomics approach, of which 204 were known metabolites and the rest were unknown metabolites. We classified these metabolites into 14 categories (Figure 1a). Among them, amino acids were the most (28%), followed by organic acids (22%), carbohydrates (22%), fatty acids (7%), nucleotides (7%), and lipids (4%), which were the main components in *R. chrysanthum* during protection against UV-B radiation.

We analyzed the difference in accumulation patterns of metabolites of different types of *R. chrysanthum* plants under UV-B treatment using a heat map (Figure 1b). The cluster heat map analysis showed that the substances in the different groups had significant differences and were divided into two clusters. The metabolites in group 1 were the highest in the control group (N) and the lowest in the UV-B domesticated *R. chrysanthum* (Q). The metabolites in group 2 were the highest in the Q group and the lowest in the N group. Different biological replicates are also clustered together, indicating good homogeneity and high reliability among biological replicates. In addition, principal component analysis (PCA) of the metabolite profiles of the 12 samples (2 × 6 biological replicates) was performed (Figure 1c). The results of PCA showed that the first principal component (PC1) explained 20.21% of the total variance, while the second principal component (PC2) explained 11.70% of the total variance. The results showed that there were great differences in metabolite accumulation patterns of *R. chrysanthumc* under different growth conditions under UV-B irradiation.

### 2.2. Differentially Accumulated Metabolite (DAM) Analysis of R. chrysanthum under UV-B Irradiation

In order to identify *R. chrysanthum* under UV-B stress between the control group and the domesticated group, multivariate statistical analyses, namely OPLS-DA, were performed; the analyses give an overview of the dynamics of the metabolome of domesticated *R. chrysanthum* under the influence of UV-B radiation (Table 1). These metabolites can be divided into 10 categories, but amino acids, carbohydrates, organic acids, and lipids dominate and are represented by boxplots. The analyses showed that in the two groups, there were mainly 23 metabolites with significant differences, among which UV-B irradiation led to a significant up-regulation of 13 metabolites in the domesticated *R. chrysanthum* (Figure 2a), and conversely, a significant down-regulation of 9 metabolites (Figure 2b).

To sort metabolic pathways based on the *p*-value involved, Metabolic Pathway Enrichment Analysis was performed. As shown in Figure 2c, metabolic pathways are involved. Metabolic pathways are shown to the right of the dashed line, *p* < 0.05. A higher *p*-value is associated with a redder color. Thus, the focus was on starch and sucrose metabolism, followed by cysteine and methionine metabolism and glycine, serine, and threonine metabolism (Figure 2c).

### 2.3. Transcriptome Analysis of R. chrysanthum under UV-B Irradiation

To further study the molecular regulation mechanisms of DAMs in the domesticated *R. chrysanthum* of *R. chrysanthum* under UV-B irradiation, RNA sequencing of UV-B domesticated *R. chrysanthum* (C1, C2, C3) and control group (B1, B2, B3) was performed. The Q20 values of all samples were 98.19–98.40%, and the Q30 values were 94.79–95.11% in the different types of *R. chrysanthum* under UV-B irradiation (Table 2). All transcriptome data are reliable for further analysis. The differential genes among the samples were screened out according to the volcanic map (Figure 3a). The number of up-regulated genes was 11,122 and the number of down-regulated genes was 6070. GO terms enriched by DEGs were analyzed to assess gene expression in domesticated *R. chrysanthum* (Figure 3a). Among the cellular component categories, the cell and cell part have a high percentage, followed by the membrane and membrane fractions. Among the biological processes, cellular processes and metabolic processes account for a high percentage. Among the molecular function categories, binding and catalytic activities accounted for the highest percentage.

### 2.4. Comprehensive Study of Metabolic Pathways of R. chrysanthum under UV-B Irradiation

To clearly understand the relationship between metabolites (carbohydrates and amino acids) and genes in *R. chrysanthum* under UV-B irradiation, we combined some metabolites and genes to build a network in order to more visually show the correlation between gene expression and metabolite accumulation. As seen in Figure 4 (Table 3), genes related to carbohydrate metabolism, HK, SUS, ENPP1-3, glgC, and GBE1 were expressed at high levels in artificially domesticated *R. chrysanthum* under UVB conditions, while WAXY was expressed at low levels. Among the genes related to amino acid metabolism, CCBL, CHA1, and MetE were slightly or significantly decreased in the UV-B domesticated *R. chrysanthum*, while CYSK was significantly increased. Carbohydrates (D-Glucose, Fructose-6-phosphate) and fatty acids (Palmitoleic acid and myristic acid) were significantly decreased in the UV-B domesticated *R. chrysanthum*, while on the contrary, amino acids (L-Cysteine, Homocysteine) and organic acids (Quinic acid) were significantly increased in the UV-B domesticated *R. chrysanthum*.

### 2.5. Physiological Changes of Domesticated R. chrysanthum under UV-B

In order to verify the role of soluble sugars and amino acids in the domestication of *R. chrysanthum* under UV-B stress, this experiment measured the changes in the content of each substance in the leaves of *R. chrysanthum*. As shown in Figure 5, the contents of amino acids in the domesticated *R. chrysanthum* were significantly increased and the soluble sugars decreased under UV-B radiation. The results showed that the domesticated *R. chrysanthum* had strong UV-B resistance.

## 3. Discussion

The degradation of the stratospheric ozone layer has led to an increase in the amount of UV-B radiation reaching the Earth and plant surfaces [14]. UV-B radiation has a large impact on many environmental factors such as plant development, growth and morphology [8,15,16]. Therefore, it is very important to study how *R. chrysanthum* responds to UV-B radiation. In this study, *R. chrysanthum* domesticated in artificial climate chambers, and those cultured in climate chambers were selected to study their adaptation to UV-B conditions. The two types of *R. chrysanthum* showed markedly different tolerances, with the domesticated *R. chrysanthum* showing marked UV-B tolerance [7]. In this study, the main metabolic components observed in relation to the resistance of *R. chrysanthum* to UV-B responses were carbohydrates and amino acids, and explored candidate genes affecting the response of *R. chrysanthum* to UV-B.

In regard to the changes in the metabolite content of these two varieties after UV-B treatment, especially the dynamic changes in amino acids and carbohydrates, osmotic regulation is a regulation method that occurs when plants are under stress [17,18]. To complete its regulation process under adverse conditions, solutes are actively accumulated in cells to reduce the osmotic potential of cell fluid to prevent excessive water loss in cells, and soluble sugar is one of the three osmotic adjustment substances, so the accumulation of sugar in *R. chrysanthum* under UV-B stress plays an important role in plant growth and ultraviolet tolerance. The accumulation of glucose plays an important role in plant growth and tolerance to ultraviolet light, and glucose is a main soluble sugar and a signaling molecule that plays an obvious central role in plant stress [19,20,21]. In this study, carbohydrate metabolism and amino acid metabolism were found to be significantly enriched (*p* < 0.05). Starch metabolism and sucrose metabolism are the main carbohydrate metabolism of *R. chrysanthum*, which takes place through two different pathways [22,23]. The first pathway occurs in the chloroplast is starch synthesis. In this process, fructose-6-phosphate produced by the Calvin cycle is converted to glucose-6-phosphate, which is then converted to starch. The second pathway involves the synthesis and breakdown of sucrose, and occurs in the cytoplasm [24]. In this process, triose phosphate is transported from the chloroplast to the cytoplasm and converted into sucrose by the action of sucrose synthase and sucrose phosphate synthase. It is well known that reduced carbon flux through sucrose synthesis in leaves may lead to reduced sugar transport and higher starch accumulation [25]. When carbohydrates are affected by environmental factors, the breakdown of starch can reduce the adverse effects caused by stress [26]. Therefore, starch accumulation and catabolism are effective responses of *R. chrysanthum* to UV-B stress [26]. Because starch is the main nutrient of *R. chrysanthum*, changes in starch metabolites affect the growth and physiological status of *R. chrysanthum*, thus affecting their quality. Carbohydrates such as starch and sucrose can be oxidized through the glycolytic pathway and the TCA cycle. Glycolysis can lead to an increase in ATP production, thereby improving the ability of plants to adapt to their environment under abiotic stress conditions [27]. The TCA cycle is essential for cellular energy production and cooperates with the carbohydrate biosynthesis pathway to maintain plant carbon homeostasis under conditions of adversity [28]. Therefore, the activation of sucrose and starch metabolism may have enabled the N group to acquire stronger UV-B resistance.

Amino acids play a major role as monomers in protein formation and are the main primary metabolites of plants. In addition, amino acids can be broken down into intermediates in many metabolic processes [29,30,31]. Amino acids play an important role in the protection of plants against different abiotic stresses and are precursors of secondary metabolites associated with plant resistance to UV-B radiation [32]. In previous studies, many amino acids, including glutamic acid, isoleucine, leucine, serine and proline, were found to accumulate in plants exposed to high levels of UV-B radiation [33]. In addition, the activity of pathways involving phenylpropanoids related to amino acid metabolism has also been shown to be increased or induced in previous studies, which in turn indirectly affect flavonoid pathways, but these results could only be detected by LC-MS [34,35]. In this experiment, we found that UV-B radiation caused significant changes in the contents of L-Cysteine, Ketoleucine, N-Acetyl-L-aspartic acid, L-Tyrosine, and Homocysteine in *R. chrysanthum*. Most amino acids were significantly increased under UV-B radiation treatment compared to the control, indicating that *R. chrysanthum* tends to accumulate amino acids when under UV-B radiation. This study found that cysteine and methionine metabolism and glycine, serine and threonine metabolism were significantly enriched in the Q group (*p* < 0.05). The cysteine content increased under UV-B radiation treatment compared to the control, indicating that cysteine responded to UV-B radiation, increased plant cell viability, protected plants from UV-B damage, and improved enzyme activity. This study found that the serine content of *R. chrysanthum* in all Q groups increased significantly with increasing stress time. Serine is an essential amino acid involved in the process of photorespiration, and its accumulation suggests that the photorespiration process of *R. chrysanthum* is enhanced under UV-B stress [34]. The increase in photorespiration eliminates excess NADPH and ATP from the cells, which is important for reduced cell damage and has positive effects.

The analyzed transcriptome data revealed that differentially expressed genes associated with UV-B stress were investigated and genes whose expression was up- and down-regulated under UV-B stress were compared, which are shown in our results. Furthermore, according to the GO analysis, among the cellular components most affected by UV-B are cells, cell parts, membranes, and membrane fractions. This suggests that plants try to prevent medium-wave UV from damaging their cells and organelles [35,36]. In addition to this, cellular and metabolic processes favor the production of antioxidants by plants in order to avoid their exposure to oxidative stress caused by UV-B radiation [37]. Binding molecular function and catalytic activity was found in the most affected genes. The difference observed for metabolize and transcripts suggest those catalytic and metabolic are likely involved in the plant’s adaptation to UV-B stress.

## 4. Materials and Methods

### 4.1. Plant Materials and Treatment

*Rhododendron chrysanthum* is also known as *Cowhide tea*, rhododendron family, *R. chrysanthum* evergreen dwarf shrub. Changbai Mountain is the place where *R. chrysanthum* grows and it is collected between 1300 and 2650 m above sea level. After transfer to the laboratory, they were placed in an artificial climate chamber (18 °C (light for 14 h)/16°C (dark for 10 h), white light irradiation of 50 μmol (photons) m^−2^ s^−1^, and 60% relative humidity) and an intelligent artificial incubator (white light irradiation of 25 °C (light for 14 h)/18 °C (dark for 10 h), and 60% relative humidity), respectively. Succession culture: the seedlings will be cultured to eight months when used for experiments. The seedlings grown in the artificial climate chamber were artificially UV-B acclimated seedlings [7]. UV-B stress was applied to seedlings of both types with an increased irradiation of 8 h per day. Undomesticated *R. chrysanthum* under UV-B radiation was used as control materials. After 2 days of UV-B stress application, leaf samples were collected and immediately frozen in liquid nitrogen. Some samples were used for RNA-seq analysis and others for GC-TOFMS analysis. For transcriptomic analyses, the mixed sampling strategy was used to eliminate the differences between individuals, and the experiment was repeated three times. Illumina Genome Analyzer deep sequencing of 6G of data was performed on each sample. For metabolomic analyses, the experiment was repeated six times. All plants were completely randomized to be used for RNA-seq and GC-MS analyses.

The artificial radiation UV-B (Philips, Ultraviolet-B TL 20W/01 RS, Amsterdam, NY, The Netherlands) used in this experiment was the same as previously described [7,37,38]. In short, the leaves of *R. chrysanthum* were exposed to UVB artificial radiation for six replicates. In the UVB treatment, a 295 nm long-pass filter (Edmund, Filter Long 2IN SQ, NJ, USA) was placed on the culture flask. The irradiance of the samples that effectively received UVB treatment was 2.3 Wm^−2^ UVB according to the transmission function of the long-pass filter, which was measured with a UV intensity meter (Sentry Optron-ICS Corp., ST-513, SHH, China) and a light meter (TES Electrical Electronic Corp., Tes-1339 Light Meter Pro., Taipei, China).

### 4.2. Determination of Physiological Characteristics of Rhododendron under UVB Stress

Six biological replicates of plant samples were frozen in liquid nitrogen. Soluble sugars and protein content were analyzed by gas chromatography, tofms (Pegasus HT, Leco, USA), gas chromatography (7890B, Agilent, Santa Clara, CA, USA), and double-headed sample MPS2 (Gerstel, Muehlheim, Germany).

### 4.3. Metabolite Identification and Quantification by GC-TOFMS

Untargeted metabolomics analysis were carried out using XploreMET platform (ver. 3.0, Metabo—profile, Shanghai, China). Sample preparation procedures and instrument setup were similar to those described previously [39]. In brief, after derivation, each sample was analyzed via the GC-TOFMS system (PEGA-sus HT, Leco, St. Joseph, MA, USA) with gas chromatograph (7890B, Agilent, USA) and a sample MPS2 with dual heads (Gerstel, Muehlheim, Germany). 

Retention index and mass spectrometry data were compared with data from the JiaLib metabolite database using XploreMET (version 3.0, Metabo-Profile) software, and metabolites were annotated. The peak area here is the relative quantitative data obtained using the chromatographic peaks integrating each metabolite.

### 4.4. Metabolite Data Analysis

The filtered data were submitted to R software (www.r-project.org (accessed on 14 April 2022)) for unsupervised principal component analysis (PCA) [40,41]. MetaboAnalyst 5.0 software (https://www.metaboanalyst.ca/ (accessed on 9 December 2021)) was used to cluster the metabolites among the samples. To identify differential metabolites, we first selected metabolites with a fold change ≥ 1.6 (upregulated) or a fold change ≤ 0.625 (downregulated) in the control group (N) compared to UV-B domesticated *R. chrysanthum* (Q). We then screened for these differential metabolites using the threshold variable importance in projection (VIP) values (VIP ≥ 1) in the OPLS-DA model. A Student’s test (mean two-tailed) (*p* = 0.05) was also used to assess the significance of differences in metabolite abundance [42], enrichment of metabolic pathways and identification of key metabolic networks using proprietary hypergeometric algorithms of XploreMET software (version 3.0, Metabo-Profile). A *p*-value < 0.05 is a significant difference and is an impact value for the importance evaluation of metabolic pathways [40].

### 4.5. RNA-Seq Library Construction and Sequencing

Total RNA processing was performed by the mRNA enrichment method. PolyA-tailed mRNA was purified using Oligo (dT) magnetic beads and Oligotex mRNA Kit. The first strand cDNA is synthesized by adding an appropriate amount of lysis agent under high temperature, and the first strand cDNA is synthesized by using the cleaved mRNA as template, then the second strand cDNA is synthesized by configuring the synthesis reaction system, and the mucous end is purified, recovered and repaired by using the kit, and the purified cDNA is ligated with an “a” at the “3” end. The product was selected according to fragment size after modification and final PCR amplification. The quality of the constructed libraries was checked by Agilent 2100 Bioanalyzer and ABI StepOnePlus Real-Time PCR system sequence after being qualified [43]. In this experiment, transcriptome sequencing was completed by the IlluminaHiSeq platform based on the Shenzhen Huada Gene Technology Research Co., Ltd.

### 4.6. De Novo Assembly and Sequence Annotation

The raw data is filtered using SOAPnuke filtering software to obtain accurate sequencing data. In this step, bases with disambiguation bases greater than 5% and low-quality bases with mass values less than 10 accounting for more than 20% of the total bases in the adapter read are considered to be low quality bases and are discarded. The remaining clean readings were assembled from scratch using Trinity software. BLASTX was used to search all transcripts with thresholds less than 10–6 in the NCBI public databases such as Nr, Nt, Pfam, KOG, and Swisspro. To annotate gene ontology terms (GO: http//www.geneontology.org (accessed on 9 July 2022)), we used the best shooting submitted to BLASTX Blast2GO program with Nr annotation. Analyses of transcript metabolic pathways were carried out using the KEGG database.

### 4.7. Differential Expression Analysis

We aligned clean reads to genomic sequences by Bowtie2, and then calculated gene expression levels for each sample by RSEM. To identify differential genes between domesticated and non-domesticated *R. chrysanthum* leaves in UVB, gene expression was calculated for each transcript using the FPKM method. To mine the genes of different *R. chrysanthum* in response to UV-B, the DEseq R package was used in the experiment to identify differentially expressed genes. In this study, significantly differentially expressed genes were screened with a false discovery rate (FDR) < 0.05, | log (fold change) | ≥ 1.

### 4.8. Statistical Analysis

Statistical analyses were conducted using R software (http://cran.r-project.org/ (accessed on 14 April 2022)) and IBM SPSS statistical software (https://www.ibm.com/analytics/spss-statistics- software (accessed on 9 March 2022)). In the results, the significance level is identified by letters. Figures were compiled using Sigmaplot 12.5 (Systa Software Inc, Chicago, IL, USA). Data were analyzed by one-way analysis of variance (ANOVA) and Pearson’s correlation tests at the 5% level.

## 5. Conclusions

In this study, we used artificially domesticated *R. chrysanthum* in a simulated alpine environment as experimental material to investigate how plants respond to strong UV radiation due to the ozone hole and to analyze the resistance of domesticated R. chrysanthum to UV-B damage by combining deep transcriptomic and metabolomic analyses, for which we constructed a pathway map in order to understand the adaptation of plants to UV-B mechanism. The potential interactions between amino acid metabolism and carbohydrate metabolism *R. chrysanthum* primary metabolism in response to UV-B radiation were identified through changes in the levels of various metabolites (amino acid metabolism and carbohydrate metabolism were mainly involved), as well as changes in individual genes associated with transcriptional stages. The results of this study deepen the understanding of the metabolic regulation of domesticated *R. chrysanthum* leaves under UV-B radiation and narrow down, as much as possible, the regulators of amino acid and carbohydrate metabolic pathways in *R. chrysanthum* under UV-B stress.

## Figures and Tables

**Figure 1 plants-11-02730-f001:**
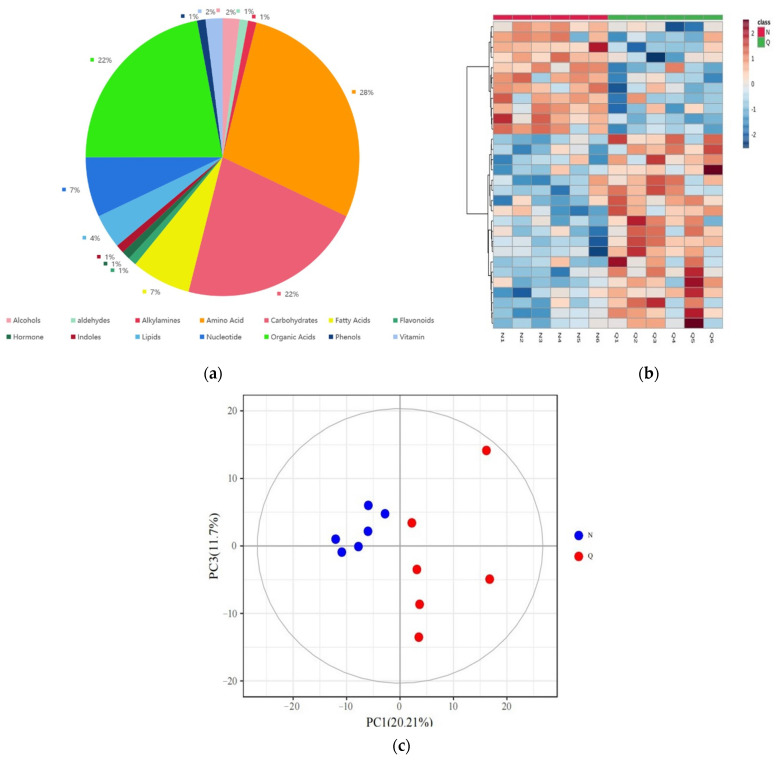
(**a**) Classification and proportion of total 404 metabolites detected in *R. chrysanthum* plants; (**b**) The heat map shows the cluster analysis of the top 30 metabolites in the control group, and the leaves of domesticated *R. chrysanthum* under UV-B irradiation. The color indicates the accumulation level of each metabolite. The redder indicates the higher the accumulation level of metabolite, while the bluer indicates the lower the accumulation level of metabolite; (**c**) Principal component analysis (PCA) of the control group and domesticated *R. chrysanthum* under UV-B irradiation.

**Figure 2 plants-11-02730-f002:**
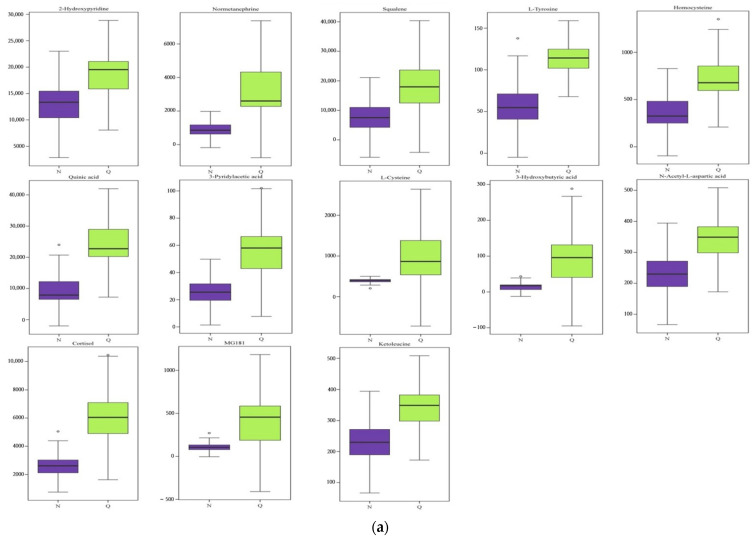
(**a**) Up-regulated metabolites of *R. chrysanthum* between control and domestication; (**b**) Down-regulated metabolites of *R. chrysanthum* between control and domestication; (**c**) Metabolic Pathway Enrichment Analysis of differentially expressed metabolites that are *R. chrysanthum* plants under UV-B irradiation.

**Figure 3 plants-11-02730-f003:**
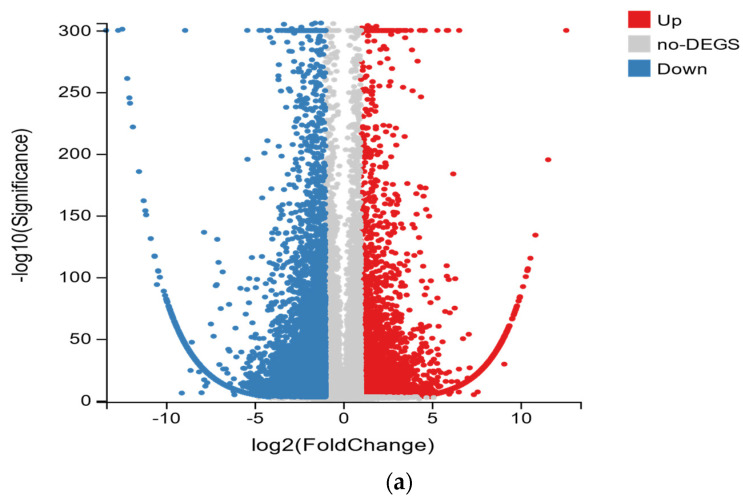
(**a**) Differential gene distribution volcano map of the control group and domesticated *R. chrysanthum*, red represents up-regulated DEGs, blue represents down-regulated DEGs, and grey represents non-DEGs; (**b**) Functional classification of DEGs based on GO, the left *Y*-axis represents GO terms, and the *X*-axis represents the number of DEGs.

**Figure 4 plants-11-02730-f004:**
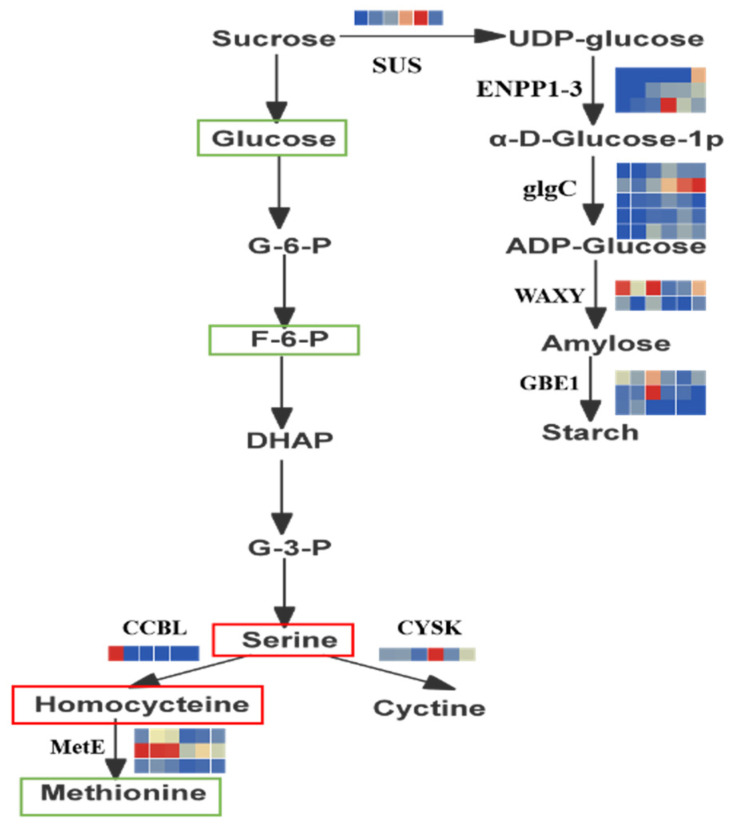
Simplified model of metabolic pathways in heatmap of expression patterns of *R. chrysanthum-related* differentially expressed genes (DEGs) under UV-B irradiation. Filtering of DEGs by log2 FPKM (transcript fragments per kilobase per million mapped reads). Blue and red boxes indicate gene down-regulation and up-regulation, respectively. HK: hexokinase; PSPH: Phosphoserine phosphatase; metE: methyltransferase; SUS: sucrose synthase; glgA: starch synthase; GBE1: 1,4-α-glucan branching enzyme; WAXY: granule-bound starch synthase; glgC: glucose-1-phosphate adenylyltransferase; ENPP1-3: ectonucleotide pyrophosphatase/phosphodiesterase family member 1/3; CysK: serine O-acetyltransferase; CCBL: cysteine-S-conjugate beta-lyase; CHA1: L-serine/L-threonine ammonia-lyase. Green and red borders indicate down-regulation and up-regulation of metabolites, respectively).

**Figure 5 plants-11-02730-f005:**
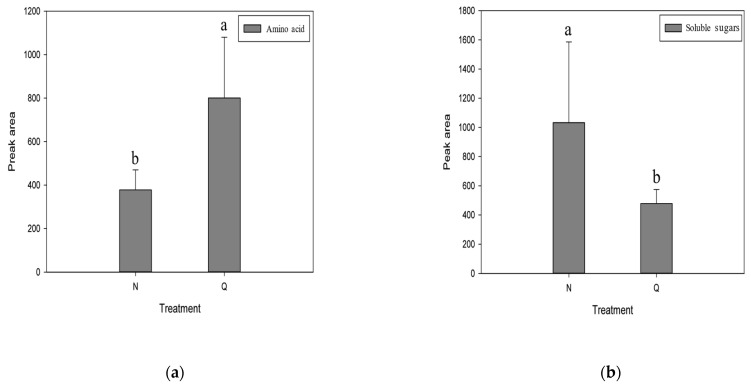
Changes in physiological indicators of *R. chrysanthum* under UV-B stress in control (N) and UV-B domestication treatments (Q). (**a**) Changes in amino acid content. (**b**) Changes in soluble sugars content.

**Table 1 plants-11-02730-t001:** Differential metabolites between domesticated and control populations were determined using multidimensional statistical analysis.

Class	Name	KEGG ID	VIP	Corr.Coeffs.	*p*
Alcohols	2-Hydroxypyridine	C02502	1.4	0.6	0.038
Alkylamines	Putrescine	C00134	1.7	−0.69	0.014
Amino Acid	L-Methionine	C00073	1.6	−0.65	0.021
	L-Cysteine	C00097	1.6	0.65	0.022
	3-Nitrotyrosine	NA	1.5	−0.64	0.026
	Ketoleucine	C00233	1.5	0.63	0.027
	N-Acetyl-L-aspartic acid	C01042	1.5	0.63	0.029
	L-Tyrosine	C00082	1.5	0.63	0.029
	Homocysteine	C00155	1.5	0.61	0.036
Carbohydrates	Ratio of D-Glucose/Sucrose	C00031/C00089	1.7	−0.7	0.012
	D-Glucose	C00031	1.7	−0.69	0.014
	Salicin	C01451	1.5	−0.64	0.026
	Galactinol	C01235	1.4	−0.57	0.055
	Fructose 6-phosphate	C00085	1.3	−0.56	0.061
Fatty Acids	Palmitoleic acid	C08362	1.7	−0.7	0.011
	Myristic acid	C06424	1.4	−0.56	0.056
Hormone	Normetanephrine	C05589	1.8	0.74	0.062
Indoles	Indoleacetic acid	C00954	1.8	−0.77	0.036
Lipids	Cortisol	C00735	1.7	0.71	0.097
	Squalene	C00751	1.6	0.66	0.02
	MG181	NA	1.6	0.64	0.024
	Phytol	C01389	1.4	−0.6	0.039
Nucleotide	Ratio of Guanine/Guanosine	C00242/C00387	1.7	−0.71	0.01
	Uridine	C00299	1.3	0.53	0.074
	Guanine	C00242	1.3	−0.52	0.082
Organic Acids	Quinic acid	C06746	1.6	0.68	0.015
	3-Pyridylacetic acid	NA	1.5	0.64	0.026
	3-Hydroxybutyric acid	C01089	1.5	0.61	0.036
	Glyceric acid	C00258	1.4	0.57	0.054
	Methylmalonic acid	C02170	1.4	0.56	0.057

Note: The correlation coefficient, representing the reliability of a particular metabolite, was obtained by correlation analysis of the scored values of the OPLS-DA model samples with the variable X (peak area of a specific metabolite in all samples). A *p*-value < 0.05 was considered significant, with a correlation coefficient < 0 indicating a significant decrease in metabolite and a correlation coefficient > 0 indicating a significant increase in metabolite.

**Table 2 plants-11-02730-t002:** Transcriptional sequence analysis of leaf samples of the control group and domesticated *R. chrysanthum*.

Sample	Total Raw Reads (M)	Total Clean Reads (M)	Total Clean Bases (G)	Clean Reads Q20 (%)	Clean Reads Q30 (%)	Clean Reads Ratio (%)
B1	50.62	42.61	6.39	98.28	94.79	84.18
B2	48.99	42.14	6.32	98.4	95.11	86.02
B3	48.99	42.11	6.32	98.35	94.99	85.96
C1	50.62	42.51	6.38	98.24	94.69	83.98
C2	50.62	42.35	6.35	98.19	94.56	83.67
C3	48.99	42.2	6.33	98.39	95.08	86.15

**Table 3 plants-11-02730-t003:** Up-regulated and down-regulated genes of *R. chrysanthum* amino acid metabolism and carbohydrates metabolism under UV-B radiation.

Category	Gene ID	Log2 (FC)	Gene Annotation	N_ FPKM	Q_ FPKM
Up-regulation	CL608.Contig2_All	2.349389594	SUS	12.91	64.39
	CL1718.Contig2_All	7.224980744	ENPP1-3	0	1.37
	CL7420.Contig1_All	2.327828344	ENPP1-3	0.38	1.8
	Unigene29857_All	3.447712663	ENPP1-3	0.32	3.34
	CL1614.Contig2_All	2.610055003	glgC	0.43	2.50
	CL1614.Contig3_All	2.062679357	glgC	2.53	10.51
	CL7994.Contig3_All	1.401056683	glgC	0.51	1.31
	CL8178.Contig2_All	2.248223204	glgC	0.33	1.49
	Unigene13491_All	1.025840993	glgC	1.34	2.65
	CL3065.Contig5_All	5.355483066	HK	0.06	2.29
	CL3235.Contig1_All	1.968881228	HK	1.36	5.14
	CL3235.Contig2_All	2.58685544	HK	1.36	7.99
	CL6896.Contig1_All	1.430008739	HK	3.90	10.42
	CL4075.Contig6_All	1.526486171	CYSK	149.10	427.61
Down-regulation	Unigene6196_All	−1.330036028	WAXY	38.85	15.33
	Unigene27695_All	−1.492972109	WAXY	9.40	3.27
	CL6569.Contig4_All	−1.344077739	GBE1	6.39	2.48
	CL6569.Contig5_All	−2.859334375	GBE1	5.58	0.76
	CL6569.Contig6_All	−8.828709849	GBE1	1.44	0
	CL2240.Contig1_All	−1.733082687	MetE	29.76	8.80
	CL2240.Contig2_All	−1.131730226	MetE	79.91	36.47
	CL2240.Contig3_All	−1.314884433	MetE	12.78	5.11
	Unigene9717_All	−4.32204255	CCBL	0.26	0
	CL1477.Contig1_All	−6.471374115	CHA1	0.80	0

## Data Availability

The data used in this study are available from the corresponding author on submission of a reasonable request.

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
