# Peer review of "UV-B Irradiation to Amino Acids and Carbohydrate Metabolism in *Rhododendron chrysanthum* Leaves by Coupling Deep Transcriptome and Metabolome Analysis"

_plants, 2022, doi:10.3390/plants11202730_

Round 1

Reviewer 1 Report

The article is well written and the research is very well prepared. Please find my comments below.

Overall, my impression was that the figure quality could be improved - but it is fine. And some can likely go to the supporting informations.

My main concern is about analysis reproducibility. None the data (mass spectrometry and transcriptomics) are deposited to reference repository. Neither are the tables used for statistical analysis. The authors used MetaboAnalyst for GC-MS analysis that has extensive output format provided for reproducibility. I would encourage sharing the summary of MetaboAnalyst analysis and the input file used. Those could supplementary informations or deposited to github or other repository.

L36 - "recovery of stratospheric ozone will take several years [5]” -> I would recommend to be more specific, like dozen, or hundreds of years ?

L50 - “unigenes” -> I am not expert, but is this really the most suited term ? This term isn’t occurring much on internet and in papers. Why unigenes rather than transcripts ?  I think at least it should be defined clearly.   L65 - "The obje tives of this study were (I) to perform parallel analysis, GC-TOFMS to quantify metabolites” This is not quantification, but measuring or analyzing. Quantification requires specific method development. Note that the authors on L71 are framing this correctly “determine the differential metabolites …”   L156 - It isn’t clear if “UV-B domesticated” refers to a variety of R. chrysanthemum plant that was naturally adapted to UV-B condition ? Or a plant that were acclimated by humans. Or a domesticated R. chrysanthemum for agricultural or ornenemental purpose. Please be more specific.   Figure 5. Please specific what is N and Q for treatment.   L186 - Seems redundant. Maybe better to bring it back at the end of discussions or conclusion.   L190 - "acclimated and non-acclimated with UV-B” see my comment above.   L193 - "In this study, combined with transcriptome data and 193 metabolic profile data, it was found that the contents of carbohydrates and amino acids 194 are the main metabolic components that affect the response of R. chrysanthum to UV-B.” I would encourage caution, as only GC-MS was performed. Other metabolites could be highly affected (even more). LC-MS would have been a nice complement for that. I think the authors could say, "the main metabolic components observed that was involved in the response of R. …”.   L208 - significantly is written twice in a row (it doesn’t make it more significant :).

L213-228 - I really like the discussion about how carbohydrates metabolism up regulation can effectively help the plant metabolism to cope with UV-B exposure.

L237 - "In addition, the 236 activity of pathways involving phenylpropanoids and flavonoids related to amino acid 237 metabolism was shown to be increased or induced [34, 35]. In this study, L-Cysteine, Ke- 238 toleucine, N-Acetyl-L-aspartic acid, L-Tyrosine, Homocysteine contents were found in 239 Rhododendron in response to UV-B radiation.” These results on phenylpropanoids pathways isn’t contradictory because those could not be seen by GC-MS, only by LC-MS. Please contextualise this explicitly when comparing your findings.

 L261- "The interactions between catalytic and metabolic processes make plant cells resistant to excessive UV-B stress, thus affecting the overall plant condition.” -> I think it would be more adapted to be less affirmative and more specific. I would use "the difference observed for metabolize and transcripts are suggesting those catalytic and metabolic are likely involved in the plant adaptation to UV-B stress etc 

L264 - “cuckoos” what is that typo ?! It seems really suspicious ...   L269 - "R. chrysanthum primary metabolism in response to UV-B radiation was revealed potential interactions.” It is nice that authors are aware that they are analyzing the primary metabolism of the plant with their GC-MS method. I think this should be specify higher in the manuscript rather than in the conclusion. Also “was revealed” seems too strong as many possible interactions are possible and couldn’t observed with the experimental design. Maybe use “has highlighted potential interactions involved in that adaptation”    Conclusion - I think the author can accentuate on the fact they leveraged a plant found both in high UV-B region and more temperate has a model to understand UV-B adaptation mechanism. This is really cool.  

Reviewer 2 Report

Congratulations

A very interesting article. For the future, it is worth carrying out research on the influence of other abiotic factors. My suggestions are in the article.

Reviewer 3 Report

In the article "UV-B irradiation to Amino acids and carbohydrate metabolism in Rhododendron Chrysanthum leaves by coupling deep transcriptome and metabolome analysis" Authors focused on important issue connected with response of R. chrysanthemum plants to UV-B irradiation. The topic of the research is interesting. Unfortunately, in my opinion the article is not suitable for publication in Plants journal. I suggest rejecting proposed article.

The main reason of my decision is the methodology used in the experiment. Authors stated that they used domesticated and undomesticated plant material. The main assumption in research is that domestic material has different strategy to deal with UV-B stress. In the Material and Methods section Authors described plant material and treatment. Unfortunately, it does not follow from the description, that one part of plants was domesticated. Also, in the text Authors write, that domesticated material was the variety of R. chrysanthemum. If so, it should by clearly stated what variety it is. Moreover, how do they know, if “domesticated” plants have different potential to deal with UV-B irradiation. 

Also, applied experimental conditions are not clear: what was the quality of white irradiance applied for the cultivation in experimental chambers? What was the source of UV-B irradiance, what was the quality and quantity of UV-B irradiance? Were plants used in experiment seedlings? It is not clearly stated what was the plant material collected in the mountains. Also, what was the source of “domesticated” material? Light intensity 50 umol/m/s is too low for effective photosynthesis in plants and light intensity in the mountains is much higher. 

Since these issues are central to the hypotheses posed by the Authors, I have decided to reject the proposed article.

With kind regards.

Reviewer 4 Report

This manuscript entitled “UV-B irradiation to Amino acids and carbohydrate metabolism in Rhododendron Chrysanthum leaves by coupling deep transcriptome and metabolome analysis” (plants-1941143) was studied the effects of UV-B irradiation on changes in metabolite contents and gene expression in Rhododendron chrysanthum grown under the domestic conditions using GC-TOFMS and RNA-Seq, respectively. The authors showed that amino acids were higher and carbohydrates were lower in domesticated plants, and transcriptome analysis further revealed that gene expression was constituted in relation to trends in sucrose and starch metabolism.
There are few detailed omics studies about UV-B effects on natural grown plants. Therefore, I feel this study is meaningful to increase in the knowledges of changes in metabolites and transcripts of R. chrysanthum under UV-B condition.

However, the organization of figures and tables needs to be improved so that readers can easily understand the results.

Minor points
1. Descriptions, numbers, and legends of each figure is too small. Please make the font size and graph larger.

2. There were several errors related to English. Please pay close attention and create your sentences carefully.

(i.e. p.11, line 298:  "2.3 W M -2" is "2.3 Wm-2" )

Round 2

Reviewer 3 Report

Dear Authors,

thank you for your response to my comments.

I recommend to show in the paper spectral composition of applied lights and UVB.

Kind regards